# Equipment Anomaly Detection for Semiconductor Manufacturing by Exploiting Unsupervised Learning from Sensory Data [note 1]

**DOI:** 10.3390/s20195650

**Published:** 2020-10-02

**Authors:** Chieh-Yu Chen, Shi-Chung Chang, Da-Yin Liao

**Affiliations:** 1Department of Electrical Engineering, National Taiwan University, Taipei 10617, Taiwan; f07921016@ntu.edu.tw; 2Straight & Up Intelligent Innovations Group Co., San Jose, CA 95113, USA; eliao@miicg.com

**Keywords:** anomaly detection, unsupervised learning, equipment sensory data, recipe-based cycle series, spectral transformation, stacked autoencoders, HDP-CVD, semiconductor manufacturing

## Abstract

In-line anomaly detection (AD) not only identifies the needs for semiconductor equipment maintenance but also indicates potential line yield problems. Prompt AD based on available equipment sensory data (ESD) facilitates proactive yield and operations management. However, ESD items are highly diversified and drastically scale up along with the increased use of sensors. Even veteran engineers lack knowledge about ESD items for automated AD. This paper presents a novel Spectral and Time Autoencoder Learning for Anomaly Detection (STALAD) framework. The design consists of four innovations: (1) identification of cycle series and spectral transformation (CSST) from ESD, (2) unsupervised learning from CSST of ESD by exploiting Stacked AutoEncoders, (3) hypothesis test for AD based on the difference between the learned normal data and the tested sample data, (4) dynamic procedure control enabling periodic and parallel learning and testing. Applications to ESD of an HDP-CVD tool demonstrate that STALAD learns normality without engineers’ prior knowledge, is tolerant to some abnormal data in training input, performs correct AD, and is efficient and adaptive for fab applications. Complementary to the current practice of using control wafer monitoring for AD, STALAD may facilitate early detection of equipment anomaly and assessment of impacts to process quality.

## 1. Introduction

In-line equipment anomaly detection (AD) identifies unusual behaviors in equipment sensory data (ESD) [1], which not only estimates the needs for equipment maintenance or repair but also affects process control for end-of-line yield. Unusual ESD behaviors could be early signals of a downgraded component, deviation of a factor or some unexpected external causes. Such anomalies impact equipment performance as well as product yield and operations management. Prompt AD facilitates proactive anomaly handling to maintain equipment performance for high product yield and productivity.

Semiconductor industry has reported that with the introduction of advanced AD, a fab could have 25% reduction in time to yield maturity, 10% increase in manufacturing capacity and 35% decrease in number of quality problems [2]. Utilizing ESD for AD has helped improve fabrication process control and raise quality [3]. Chien et al. research [4] indicated that with applications of big data analytics by exploiting various data sources in semiconductor fabrication, troubleshooting time can be effectively reduced and the production yield largely increased. Chen et al. proposed in [5] a yield alert and diagnostic analysis framework and showed that it helps quickly clarify the causes of abnormal product yield. Prompt equipment AD exploiting ESD leads to proactive equipment maintenance and process control and is of significant value to yield management of an operational fab.

AD of semiconductor tools which fabricate layers of circuitry onto silicon wafers is very challenging because of tool diversity and process complexity. ESD items of a tool are also highly diversified, which may involve hundreds of items, and drastically scale up along with the increased use of sensors [6]. Even veteran engineers lack knowledge about individual ESD items. Control wafer monitoring is a common approach to measure wafer processing effects by a tool and to evaluate equipment conditions and performance. Control wafer monitoring incurs high costs of control wafers and their processing, tool capacity losses from processing control wafers, and time delay and costs of measuring control wafers. Exploiting ESD for equipment AD can provide timely and cost-efficient identification of equipment anomalies and warnings on potential yield and productivity losses. However, with the increasing use of sensors, the growth of equipment and process complexities, and the high ESD sample rates, it has become formidable for equipment engineers to comprehensively grasp detailed characteristics of all kinds of ESD.

In semiconductor fabrication, the fault detection and classification (FDC) function is a common practice to collect ESD and organize collected data into a database. ESD consists of many sensory data items acquired from sensors of each tool. A sensory data item collected from a tool sensor is also called as a Status Variable Identification (SVID), for example, gas flow, power, pressure, current, voltage, temperature, position, etc. In general, FDC periodically collects data of an SVID from the corresponding tool, for example, one or ten data samples per second. During the processing of a wafer, FDC acquires ESD of hundreds of SVIDs from built-in and add-on sensors of each tool [7]. There are tens of tool groups with various functionality and tens of tools in each tool group. Therefore, ESD data rate to FDC is millions of data samples per second. Both historical and real-time ESD are stored in the form of time series in FDC for further AD analyses.

ESD is useful for monitoring the condition of a tool and for inferring how the tool performs in processing wafers. A semiconductor fabrication process consists of hundreds of processing steps. Each processing step requires processing by a tool of specific functionality including deposition, removal, patterning, and modification of electrical properties [8]. The set of tool instructions which specify how a processing step is to be performed is called a *recipe* [9]. The time a tool takes to process a recipe on a wafer is a *cycle* since the processing is repetitive for wafers of the same product. The ESD of an SVID collected by FDC in a cycle forms a *sample sequence*, which is a time-stamped series of sampled sensory data values from the SVID. Sample sequences of one SVID resulted from processing one same recipe by a tool should have very similar patterns or features when the tool is normal. Although sample sequences of various SVIDs differ in time series patterns or features, they are generated during tool processing of one same recipe. Thus sample sequences are highly correlated among SVIDs of the tool. Equipment engineers can monitor sample sequences from various SVIDs of individual tools for anomaly detection. However, semiconductor tool complexity and ESD data volume and rates are so high that an engineer can easily be overwhelmed.

Take a high density plasma chemical vapor deposition (HDP-CVD) tool for example [10]. It performs an enhanced CVD gap-fill process. An HDP-CVD recipe basically consists of wafer allocation, chamber evacuation, plasma generation, electric bias construction, ion bombardment, and end recovery. High density plasma is generated in the chamber with very low chamber pressure. Electric bias forces ions in plasma to bombard the wafer and leave reactants on the surface which fills the gap. Sophisticated physics involve plasma, electric field, chemical reactants, and their interactions. Typically, there are more than 400 SVIDs and the sampling time of an SVID is 1 s. So, the ESD rate is more than 400 samples per second per tool. Sophisticate physics of an HDP-CVD tool further complicate its anomaly detection.

There has been research work on general equipment AD by using ESD in time series [11] and some on semiconductor manufacturing tools in specific [12]. Machine learning-based methods have been popular solutions. Fan and Hsu [13] proposed an equipment AD framework which first identifies key SVIDs, and then trains multiple classifiers for determining if sample ESD is abnormal. Iqbal et al. [14] applied a moving window principal component analysis model with recipe information to capture the normal model of ESD and detect potential tool anomalies. Liu et al. [15] exploited knowledge about event ordering relationship to structure neural networks, improve equipment anomaly detection accuracy and reduce computational complexity. For time series data, Hsu and Liu [16] proposed a multiple time-series convolutional neural network model trained by labeled normal and abnormal ESD for equipment AD. To identify the normality and/or anomaly, all these AD methods require sufficient prior knowledge about ESD used and the corresponding tool such as labeled normal and abnormal ESD, recipe information related to ESD features, and the ordering of tool events.

Some state-of-the-art studies of equipment AD in other problem domains than semiconductor manufacturing do not require much prior knowledge. To minimize the requirement of prior knowledge, machine learning techniques, especially autoencoder-based architectures, are widely applied to equipment AD for machines in various domains. Borghesi et al. [17] adopted a set of general autoencoders to learn normal behaviors of high performance computing systems and exploits learned autoencoders to identify abnormal behaviors. Oh and Yun [18] specifically used the residual error of a general autoencoder to identify the anomaly in a surface-mounted device (SMD) machine. Kim et al. [19] applied a variational autoencoder to capture the distribution of healthy state data of a linear motion guide. Park and Yun [20] proposed an AD scheme using recurrent neural network encoder-decoder that significantly speeds up computation for fast anomaly detection of an SMD machine. These methods efficiently capture normality by training on normal ESD, which may often be difficult to obtain.

Spectral analysis-based AD approaches are also commonly adopted for equipment involving vibrations, rotating, or periodic signals [21], and have achieved successes in various domains. Taylor et al. [22] applied frequency transformation and analyzed frequency features of network data for AD in the automotive controller area network bus. Liu et al. [23] developed a new AD algorithm that combines time-frequency domain analysis and artificial intelligence technology for detecting abnormal vibrations in a large power transformer. Rajendran et al. [24] exploited power spectral density data to detect abnormal behavior in wireless spectrum. Frequency components of a vibration signal are often useful for anomaly analysis of a vibrating or rotating machinery [25,26]. In semiconductor equipment AD, there have been only few spectral-analysis related approaches. Liao et al. [27] considered the spectral transformation of ESD cycles and preliminarily investigated the value of spectral analysis for semiconductor AD. Chen et al. [28] further indicated how spectral analysis helps detect a drift in ESD anomaly at very low frequencies.

Liao et al. [27] assessed the feasibility of using a Stacked AutoEncoder (SAE) for unsupervised learning of normal features for AD from semiconductor tool ESD. An SAE contains multiple hidden layers of nodes in addition to the input and output layers and can be viewed as the “stacking” of multiple general autoencoders [29]. Preliminary results show that an SAE is capable of learning both the temporal and spectral features of ESD among cycles of each recipe [28] and that a simple test on the difference between the learned features and the tested achieves AD over a small data set.

The Numenta Anomaly Benchmark (NAB) [30] is an open-source framework for evaluating the unsupervised real-time anomaly detection approaches. Some public datasets and benchmarks are provided in NAB. However, these datasets do not contain data with characteristics similar to semiconductor ESD.

In this paper, on top of the preliminary design and insights achieved by Liao et al. [27] and Chen, Chang and Liao [28], we shall further define four design problems of semiconductor equipment AD exploiting ESD. To address the four problems and resolve their challenges, we shall propose a novel framework of spectral and time autoencoder learning for anomaly detection (STALAD). STALAD design has four innovations: (1) identification of cycle series and spectral transformation (CSST) from ESD, (2) unsupervised learning from CSST of ESD by exploiting Stacked AutoEncoders, (3) hypothesis test for AD based on the difference between the learned normal data and the tested sample data, (4) dynamic procedure control enabling periodic and parallel learning and testing. Applications to ESD of an HDP-CVD tool demonstrate that STALAD learns normality without engineers’ prior knowledge, is tolerant to up to 30% of abnormal data in training input, performs correct AD, and is efficient and adaptive for fab applications. Complementary to the current practice of using control wafer monitoring for AD, STALAD may facilitate early detection of equipment anomaly and give warning to yield management up to two weeks in advance according to our case study.

The remainder of this paper is organized as follows: Section 2 describes the equipment anomaly detection problems and challenges. Section 3 presents the framework design and operation procedure of the two-phased STALAD. Section 4 is the detailed design of learning phase, where STALAD learns normality from ESD and outputs learned features. Section 5 describes the testing phase design, where STALAD detects anomaly based on the difference from the learned normality and a hypothesis test. Section 6 evaluates the performance of STALAD with case studies by using ESD collected from an HDP-CVD tool. Finally, Section 7 concludes the paper.

## 2. Anomaly Detection Problems and Challenges

Abundant equipment sensory data (ESD) is ready for use in most of the modern semiconductor fabs. As has been described in the Introduction, making effective use of ESD for AD is challenging. This section will describe ESD characteristics with ESD of an HDP-CVD tool as an example, define the problems and describe the challenges of exploiting ESD for equipment AD.

### 2.1. ESD Characteristics

A *recipe-based cycle*, called a *cycle* hereafter, is the time period of ESD collected from processing a specific recipe over a wafer by a tool. A *sample sequence* is a time-stamped series of sampled sensory data values from the SVID of a cycle. Figure 1 shows an exemplary sample sequence from an SVID collected from an HDP-CVD tool, where the sampling rate is one sample per second and the cycle consists of 189 time samples. In semiconductor manufacturing, most of ESD time series when plotted over time axis approximately show piecewise linear value curves [31]. For example, the sample sequence of the cycle depicted by Figure 1 starts at value 250, rises to a range between 325 and 375 after 30 s, and drops back to 250 at 180 s [27,28].

This paper further defines the notion of *recipe-based cycle series (CS)*, which is composed of sample sequences of some concatenated sequential cycles of an SVID. Figure 2 gives exemplary CSes from 3 different SVIDs collected synchronously from an HDP-CVD tool processing over 10 wafers. The three CSes are composed of 10 continuous cycles each. Since each cycle contains 189 time samples, one sample per second, Figure 2 depicts the three CSes over 1890 s. Although the three CSes belong to 3 different SVIDs, they all show the same and apparent periodicity because of the repetitive processing of one same recipe by the tool.

A *normal sequence* is a sample sequence representative of all sample sequences of cycles in a CS. Figure 3 depicts a CS of 4 cycles. The sample sequences of the first three cycles are quite similar and variations among them are small. The variations may be caused by (1) processing noises, (2) switching from one recipe to another and equipment tuning or maintenance [7], and (3) sensing noises. With existence of these variations, even veteran engineers often cannot easily identify whether a sample sequence can be considered as a normal sequence.

An *abnormal sequence* is a sample sequence which significantly differs or deviates from the normal sequence. How much difference is significant depends on the characteristics of an SVID and requires a careful analysis. For example, in Figure 3, the beginning part of the sample sequence of the 4th cycle has a slightly upward trend, which differs from the downward pattern in the other three sample sequences. Significant differences may be rooted in (1) failures of equipment parts, (2) malfunction of sensors, (3) malicious activities by intruders, or even combinations of the above. Since the difference between normal variations and abnormal deviations may very often be subtle, anomaly is very challenging when AD is further complicated by ESD generation in very high volume and rates [28]. In general, even veteran engineers may not have sufficient knowledge to identify whether a sample sequence is abnormal.

### 2.2. Problems and Challenges

To achieve early AD, we exploit the available ESD in FDC, as characterized in Section 2.1. There are four research problems, each corresponding to a design issue. Problem definitions and descriptions of their respective challenges are as follows:


***P1. Exploitation of characteristics of CSes with limited knowledge about ESD***


With limited knowledge about ESD, how do we make good exploitations of characteristics of CSes to support equipment AD for semiconductor tools?


***Challenge 1***


Characteristics of CSes from different SVIDs are diversified and complex and vary with tools. There is little known knowledge about ESD that can be directly applied for anomaly detection.


***P2. Learning of normal features from a CS possibly containing some abnormal sequences***


How do we design a methodology to efficiently learn normal features from a CS that may contain unknown amount of abnormal sequences?


***Challenge 2***


Input data may consist of some abnormal sequences. Learned features are thus not guaranteed to be normal features.


***P3. Anomaly detection in a sample sequence by learned normal features***


With learned normal features, how do we design a test to decide whether a sample sequence is abnormal?


***Challenge 3***


Measuring the deviation of a sample sequence from the representative sequence with the learned normal features is difficult because normality may not be fully described by learned normal features only. The other difficulty is to quantitatively define the deviation of two sample sequences. How much deviation normality will a sample sequence be considered as abnormal is also challenging.


***P4. Fast and adaptive detection for proactive yield improvement***


How do we make AD efficient in computation and adaptive to fab status so that early AD by using ESD may contribute to proactive yield and productivity improvements?


***Challenge 4***


High ESD volume and rates pose the need for a time-efficient detection methodology. Moreover, the change of the tool normality over time as fab status evolves restricts the applicability of the learned normality features; they have to be updated from time to time but when?

## 3. STALAD Framework Design

This section proposes an innovative design for AD, the Spectral and Time Autoencoder Learning for Anomaly Detection (STALAD) framework, which exploits unsupervised learning techniques to learn features of sample sequences and test anomalies for individual SVIDs. The STALAD framework has four innovations:I1.Identification of cycle series hidden in ESD and spectral transformation of cycle series,I2.Unsupervised learning of normal features from cycle series and their spectral transformation by exploiting Stacked AutoEncoders,I3.One-tail hypothesis test for anomaly detection based on the difference between the learned normal sequence and the tested sample sequence, andI4.Dynamic procedure control which coordinates both learning and testing phases to enable periodic and parallel executions of learning and testing.

These innovations in STALAD provide solutions to challenges 1–4 of problems P1-P4 described in Section 2.

### 3.1. Framework Overview

Figure 4 depict the STALAD framework and its interactions with the FDC database and data cleansing modules. The FDC database constantly collects ESD from individual tools and stores it in a time series form for further access. When STALAD requests for data, the FDC database will pass the requested data to a data cleansing module first. A data cleansing module cleanses data by some presetting rules automatically with engineers’ review and some manual adjustments. The cleansed data is input to STALAD.

There are two phases in STALAD: (1) the unsupervised normal feature learning phase (*Learning phase* for short) that requires a batch of cleansed historical data, and (2) the real-time feature testing phase (*Testing phase* for short) that requires cleansed per-wafer data. An AD procedure controller controls operations of the two phases. In more details, STALAD consists of six functional modules as shown in Figure 4, where the four novel modules corresponding to the four innovations are marked in red. Either phase contains a data preprocessing module, which identifies CSes hidden in the input ESD, computes spectral transformation (ST) of sample sequences of CSes, and outputs sample sequences of CSes and their STs together, denoted as *CSSTs*. In the Learning phase, the unsupervised normal feature learning module learns normal features, by exploiting SAEs, from the preprocessed historical CSST, and outputs learned SAEs weights and corresponding test thresholds. In Testing phase, the time/frequency SAE-based feature testing module takes the learned SAEs weights and corresponding test thresholds as inputs, performs one-tail hypothesis test on the preprocessed test data of per wafer CSST, and outputs per wafer test result labeling as an AD result of the tested wafer. An equipment health dashboard collects per wafer test labels and visually displays them in lights for engineers’ further analyses.

The AD procedure controller starts with notifying Learning phase and Testing phase, and ends with the finish signals sent from Learning phase and Testing phase, respectively. Figure 4 also depicts all control signal flows and data flows among modules. In the figure, a solid arrow line represents a control signal flow and its direction, and a dash arrow line represents a data flow and its direction. The pairs of learned SAE weights and test thresholds of all SVIDs are defined as the *learned normality*. Once a learned normality is computed, it will be transmitted to Testing phase by the AD procedure controller to detect anomalies by the latest learned normality. The procedure design is based on the following four principles:Batch processing in the Learning phase: To ensure the success of normality learning, sample sequences of ESD from a batch of many processed wafers are used for training.Per wafer processing in the Testing phase: To detect as early as possible, once available from the tool, per wafer sequence is used for testing against the learned normality.Regular activation of Learning phase: To deal with the change of the normality of the tool, the Learning phase will be activated regularly after a period of time, which is typically a week, to learn the new normality of the tool.Parallel execution between phases: To handle the high-rate coming of ESD, the Testing phase keeps the previous learned normality so that it can remain per wafer testing while the Learning phase is computing the new learned normality.

The detailed procedure controls consist of three stages as follows:


Initialization and training stage: having no learned normality
(1)The AD procedure controller activates the Learning phase when the STALAD begins to run.(2)The data-preprocessing-into-CSST (batch wafers) module queries a batch of historical ESD from the FDC database.(3)The FDC database responds with the batch of historical ESD and sends the batch to data cleansing and then to the data-preprocessing-into-CSST (batch wafers) module.(4)After preprocessing, the data-preprocessing-into-CSST (batch wafers) module sends the batch of CSSTs to the unsupervised-normal-feature-learning module for training.(5)After training, the unsupervised-normal-feature-learning module sends a finish signal together with the learned normality to the AD procedure controller.



Testing stage: having a learned normality
(6)When a learned normality is available from Learning phase, the AD procedure controller activates the Testing phase, and sends the learned normality to the time/frequency SAE-based feature testing module.(7)The data-preprocessing-into-CSST (per wafer) module queries the FDC database for per wafer ESD just generated by a tool.(8)The FDC database responds with the requested per wafer ESD and sends the per wafer data to cleansing and then to the data-preprocessing-into-CSST (per wafer) module.(9)After preprocessing, the data-preprocessing-into-CSST (per wafer) module sends the converted per wafer test CSSTs to the time/frequency-SAE-based-feature-testing module for testing.(10)After testing, the time/frequency SAE-based feature testing module sends a finish signal to the procedure controller and displays the test results on the equipment health dashboard. The equipment health dashboard displays detection results of all SVIDs by a list of lights indicating normal or abnormal.(11)Once the next per wafer ESD comes, the AD procedure controller activates the Testing phase again, repeat steps 6 to 10 using the current available learned normality to test it.



Update stage: updating the learned normality
(12)If an anomaly is detected and the corresponding tool maintained, or if equipment engineers notify tool adjustments, the AD procedure controller will activate the Learning phase to restart learning, and repeat steps 2 to 5 for learning the new normality.


The following Section and Section 4 and Section 5 will present detailed designs of STALAD. To provide quantitative descriptions, let us first define some notations to be used. Table 1 lists notations of constants and given sets. Table 2 defines a few functions. Table 3 denotes indices. Table 4 lists variable and vector notations. For a test threshold, learned SAE weights, and the learned normality, we may omit the superscript t or s in their expressions if they are applicable for both time series and spectra. Descriptions of AD procedure in STALAD will be focused on AD of single SVID, say SVID j. The same procedures apply to all SVIDs.

### 3.2. Data Preprocessing into Cycle Series and Spectral Transformation

In STALAD, there are two data preprocessing modules–one for the Learning phase of batch wafer data, and the other for the Testing phase of per wafer data. The major difference between these two parts is the number of cycles in its output CS, while their preprocessing procedures are the same. In this Section, only the preprocessing design for batch wafer data is described.

The apparent periodicity shown in a CS inspires two innovative designs of the data preprocessing module: (1) identifications of CSes hidden in ESD, and (2) spectral transformation of sample sequences in a CS to characterize its periodic features. Since the spectral transformation has been proven an effective way to analyze a periodic signal [22], we also exploit spectral transformation and take spectral features of CSes into consideration.

Figure 5 depicts the data preprocessing module for each SVID in STALAD. The following descriptions focus on ESD from only single SVID j, single recipe, and single tool. It takes cleansed ESD $\left\{y_{njk},\;n=1,\ldots ,N,\;k=1,\ldots ,K\right\}$ as input and outputs a CSST $\left(y_{j},{\tilde{\nu }}_{j}\right)$. First, the cycle identification submodule groups cleansed ESD into all sample sequence $\left\{y_{nj},\;n=1,\ldots ,N\right\}$. Each sample sequence $y_{nj}$ is a vector composed of ordered ESD of the wafer n:(1)$$y_{nj}=\left[y_{nj1},y_{nj2},\ldots ,y_{njK}\right]$$

The CS concatenation submodule concatenates all sample sequences into a CS $y_{j}$ by the ordering of processing wafers:(2)$$y_{j}=\left(y_{1j},y_{2j},\ldots ,y_{Nj}\right)$$

Each sample sequence in the CS is transformed into a magnitude spectrum by the spectral transformation submodule performing FFT. That is, given $y_{nj}$, its magnitude spectrum $\nu_{nj}$ is:(3)$$\nu_{nj}=\left[\nu_{nj1},\nu_{nj2},\ldots ,\nu_{njK}\right],\;\mathrm{where}\;\nu_{njk}=\left|\sum_{m=1}^{K}y_{njm}e^{-i\frac{2\pi }{K}km}\right|.$$

Because certain symmetries in the magnitude spectrum exist for a real-valued $y_{nj}$, the one-side magnitude spectrum is sufficient to express the magnitude spectrum [32]. The spectral transformation submodule also computes the one side magnitude spectrum ${\tilde{\nu }}_{nj}$ of $\nu_{nj}$:(4)$${\tilde{\nu }}_{nj}=\left[\nu_{nj1},\nu_{nj2},\ldots ,\nu_{nj\tilde{K}}\right],\;\mathrm{where}\;\tilde{K}=h\left(K\right)$$

Finally, the spectral transformation submodule collects all magnitude spectra as a collection ${\tilde{\nu }}_{j}$,
(5)$${\tilde{\nu }}_{j}=\left({\tilde{\nu }}_{1j},{\tilde{\nu }}_{2j},\ldots ,{\tilde{\nu }}_{Nj}\right),$$
and outputs a CSST $\left(y_{j},{\tilde{\nu }}_{j}\right)$, the pair of the CS and all magnitude spectra with respect to the SVID j.

## 4. Unsupervised Feature Learning and Validation

We design the unsupervised normal feature learning phase in STALAD to find normal features from CSSTs of individual SVIDs. SAE-based feature learning makes it able to perform unsupervised feature learning for all kinds of SVIDs. Learning phase is initiated by the AD procedure controller, takes historical ESD as input, and outputs the learned normality of each SVID. Learning phase exploits SAEs for unsupervised learning of the salient features of CSSTs. It also leaves part of input historical ESD as validation data to generate test thresholds for Testing phase. Functionality evaluation suggests effective normality learning by the learning phase design.

### 4.1. Normal Feature Learning Design

Figure 6 depicts the unsupervised normal feature learning module which consists of 3 submodules: (1) data splitter, (2) SAE training, and (3) threshold generator. The module takes CSSTs $\left(y_{j},{\tilde{\nu }}_{j}\right)$ of SVID j as input. The modules outputs learned SAE weights $W_{j}^{t}\;\mathrm{and}\;\;\;W_{j}^{s}$ and test thresholds $\theta_{j}^{t}\;\mathrm{and}\;\theta_{j}^{s}$ for time series and spectra respectively. Namely, the module outputs the learned normality $\left(\theta_{j}^{t},W_{j}^{t}\right)$ for time series and $\left(\theta_{j}^{s},W_{j}^{s}\right)$ for spectra.

The data splitter splits CSSTs $\left(y_{j},{\tilde{\nu }}_{j}\right)$ into two parts by their ordering of cycles. Let r be a user-specified ratio. Suppose a CSST contains N cycles, the former ⎣rN⎦ cycles are split into training data $D_{train}=\left\{\left(y_{nj},{\tilde{\nu }}_{nj}\right),n=1,\ldots ,⎣rN⎦\right\}$ for the SAE training, and the latter N−⎣rN⎦ cycles are split into validation data $D_{valid}=\left\{\left(y_{nj},{\tilde{\nu }}_{nj}\right),n=⎣rN⎦+1,\ldots ,N\right\}$ for the threshold generator. We suggest r to be around 0.75, which follows the tradition of splitting training and validation data [33].

The SAE training submodule contains one SAE ft for learning from time series data ynj and the other SAE fs for learning from spectra input ν˜nj. We exploit the property of SAEs [34] to learn normal features of sample sequences in Dtrain. Let len(ynj)=K. The SAE for time series ynj consists of 2M+1 fully-connected layers sequentially with dimensions K, h(K), …, hM−1(K), hM(K), hM−1(K), …, h(K), and K, where:(6)M≜min{M∈ℕ|hM(K)≤β}
and β is a user-defined maximum number of the learned features. Therefore, the SAE for time series ynj contains 2M+1 layers with the former M+1 layers being the encoder part, and the latter M+1 layers being the decoder part. Each layer in the encoder part halves the dimension. Equation (6) tells how many dimensional reduction layers are needed. We suggest 5≤β≤10 so that the number of the learned features are enough to characterize a CSST well. According to Equation (4), len(ν˜nj)=h(K). The SAE for frequency spectra ν˜nj consists of 2M−1 layers with dimensions h(K), h2(K), …, hM−1(K), hM(K), hM−1(K), …, h2(K), h(K).

All cycles in data set $D_{train}$ are used as training samples to train SAEs. We define the n-th *decoded sequence* outputted by the SAEs as $\left(y_{nj}^{\prime },{\tilde{\nu }}_{nj}^{\prime }\right)$, where:(7)$$y_{nj}^{\prime }≜f^{t}\left(y_{nj};W_{j}^{t}\right)$$
(8)$${\tilde{\nu }}_{nj}^{\prime }≜f^{s}\left({\tilde{\nu }}_{nj};W_{j}^{s}\right)$$

We define the *difference value* of the n-th cycle as the mean square error between the input sample sequence and its decoded sequence:(9)$$d_{nj}^{t}≜\frac{1}{len\left(y_{nj}\right)}||y_{nj}-y_{nj}^{\prime }||{}^{2},\;\mathrm{and}$$
(10)$$d_{nj}^{s}≜\frac{1}{len\left({\tilde{\nu }}_{nj}\right)}||{\tilde{\nu }}_{nj}-{\tilde{\nu }}_{nj}^{\prime }||{}^{2}.\;$$

We define the loss function of the SAE as the sum of difference values of all cycles in $D_{train}$. We minimize SAE loss by the Adam optimization method with its suggested settings [35]. The training iteration terminates when exceeding a predefined maximum iteration number, which is set around 10,000 to prevent the SAE from overfitting.

The threshold generator submodule determines the test thresholds based on the learned SAE $W_{j}$ and the validation data set $D_{valid}$. The validation difference values over $D_{valid}$ have sample mean $\mu_{j}$ and sample variance $\sigma_{j}^{2}$ as follows:(11)$$\mu_{j}=\frac{1}{N-⎣rN⎦}\overset{N}{\underset{n=⎣rN⎦+1}{\sum }}d_{nj}$$
(12)$$\sigma_{j}^{2}=\frac{1}{N-⎣rN⎦-1}\left(\overset{N}{\underset{n=⎣rN⎦+1}{\sum }}d_{nj}^{2}\right)-\mu_{j}^{2}$$

Since the difference value is essentially the sample mean of $\left\{{\left(y_{njk}-y_{njk}^{\prime }\right)}^{2},k=1,\ldots ,K\right\}$, we assume that difference values of normal sequences follow a normal distribution according to the central limit theorem [36]. Therefore, we define the test threshold $\theta_{j}$ as the following:(13)$$\theta_{j}≜\mu_{j}+\alpha \sigma_{j}$$

Typically α is chosen to be 3, while the larger the α is, the looser the threshold of normality is. The submodule pairs the learned SAE weights $W_{j}$ with the test thresholds $\theta_{j}$ of individual SVIDs, and outputs the learned normality $\left(\theta_{j},W_{j}\right)$.

### 4.2. Functionality Evaluation

Two functionalities of the learning phase should be evaluated: (1) computational time, and (2) representativeness of the learned normality—how much representativeness a sample sequence has to be a normal sequence. A simple HDP-CVD anomaly case is considered, where veteran engineers can easily identify normal and abnormal sequences. Computational time is measured, and the learned normality is validated by engineers.

#### 4.2.1. Evaluation Dataset

The dataset contains 97 cycles collected from 8 SVIDs of an HDP-CVD tool. Each cycle lasts for 218 s and therefore has 218 data points. The first 58 cycles are used as training data, the middle 19 cycles as validation data, and the last 20 cycles as testing data.

#### 4.2.2. Settings

We run STALAD using Python on a Windows^TM^ (Microsoft, Redmond, WA, USA) platform, and train SAEs with the help of an NVIDIA GTX 1060 GPU^TM^ (NVIDIA, Santa Clara, CA, USA) using the TensorFlow^TM^ (Google Brain Team, Mountain View, CA, USA) module. The computational time is measured using the “time” built-in Python module. The hyperparameter settings of SAEs are summarized in Table 5.

#### 4.2.3. Evaluation Metrics

Let $d_{tj}$ be a difference value of a testing cycle t. Define the *identification factor*
$\alpha_{tj}$ to the cycle t as the following:(14)$$\alpha_{tj}≜\frac{d_{tj}-\mu_{j}}{\sigma_{j}}$$
where $\mu_{j}$ and $\sigma_{j}$ are calculated by using Equations (11) and (12). A small $\alpha_{tj}$ value means that the sample sequence is likely to be considered normal, and vice versa.

#### 4.2.4. Expected Results

Due to relatively deep architecture of an SAE and ten thousand iterations for the training procedure, a few minutes are expected to train an SAE. We expect a STALAD time SAE or spectral SAE can learn normal sequences due to the dominant amount of normal sequences in the training data which contains 57 normal sequences and only one abnormal sequence. If we pick a normal sequence in the test data, we expect $\alpha_{tj}<\alpha$, which means this normal sequence is similar to a normal sequence.

#### 4.2.5. Evaluation Results and Discussions

The computational time of the learning phase on 58 cycles of the 7th SVID in the training data is 47.966 s for time series input and 40.453 s for frequency spectra input. Time series learning takes more time for training since its SAE has two more layers than a spectral SAE. Since the idle time between two adjacent wafer cycles is generally more than 1 min, the learning phase, which takes only 47.966 s, is able to finish before the processing of the next wafer. For the case that the batch data for training is 58 wafer cycles, the learned normality should be ready for AD on the next wafer.

Figure 7a depicts the difference value for a normal sequence in time series. The left part shows the normal sequence input (in blue) only and the right part shows the SAE decoded sequence (in orange) as well as the normal sequence. As the sensory values range from 2500 to 6000 and the difference between two sensory values on a sample time point is generally less than 100, little difference can be identified visually on the figure. The difference value is $d_{tj}=480.200$, and $\mu_{j}=1276.024$, $\sigma_{j}=1732.268$. The relative magnitude of difference $\left|\alpha_{tj}\right|$ is about 0.459, which is much smaller than 3. This demonstrates that Learning phase successfully learns the normal sequence.

Figure 7b depicts the difference value for the normal sequence in frequency spectra. The difference value is $d_{tj}=6.186$, and $\mu_{j}=10.177$, $\sigma_{j}=16.427$. Thus $\left|\alpha_{tj}\right|$ is about 0.243, which is also much smaller than 3. Again, this demonstrates that the learning phase successfully learns the spectrum of the normal sequence.

## 5. Feature Testing for Real-time Anomaly Detection

Once the normality features of each SVID are available by using the SAE-based learning, the AD procedure controller then runs the testing phase of STALAD. Whenever a tool completes processing a recipe over a wafer, the testing phase takes the preprocessed ESD of individual SVIDs of the wafer as input and tests if the SVID data sequence of the recipe cycle is close to the learned normality or not. The test phase design consists of SAE-based encoding and decoding of the input data for comparison with the learned normality and a one-tail hypothesis test to determine if the difference is significant. Detailed descriptions of the designs are as follows.

### 5.1. SAE-Based Feature Testing Design

Figure 8 depicts the time/frequency SAE-based feature testing module. Inputs to the module are the learned normality of individual SVIDs $\left\{\left(\theta_{j},W_{j}\right),\;j=1,\ldots ,J\right\}$ and the CSSTs $\left\{\left(y_{tj},{\tilde{\nu }}_{tj}\right),\;j=1,\ldots ,J\right\}$ of a wafer for test, where the subscript t indicates data to test. The module outputs AD test labels of the wafer. The module is composed of SAE testing and one-tail hypothesis test submodules to determine if the CSSTs $\left\{\left(y_{tj},{\tilde{\nu }}_{tj}\right),\;j=1,\ldots ,J\right\}$ of the wafer for test has a significant difference with the learned normality.

The SAE testing submodule exploits learned SAEs to encode and decode the CSSTs for test. It accepts CSSTs $\left\{\left(y_{tj},{\tilde{\nu }}_{tj}\right)\right\}$ of a wafer for test and SAE weights $\left\{W_{j}\right\}$ of the learned normality. The decoded sequences $y_{tj}^{\prime }$ and ${\tilde{\nu }}_{tj}^{\prime }$ are computed according to Equations (7) and (8). Then it outputs the *decoded test CSSTs,*
$\left\{\left(y_{tj}^{\prime },{\tilde{\nu }}_{tj}^{\prime }\right)\right\}$, where each pair consists of the SAE-decoded time series and the frequency spectral sequences. The submodule keeps SAE weights $\left\{W_{j}\right\}$ of the learned normality for testing *CSSTs* of the next wafer until the learning phase updates the learned normality.

The one-tail hypothesis test submodule accepts as inputs $\left\{\left(y_{tj},{\tilde{\nu }}_{tj}\right)\right\}$, the decoded test CSSTs of a wafer. The submodule then computes the difference values $d_{tj}^{t}$ and $d_{tj}^{s}$ by using Equations (9) and (10) and the test thresholds $\left\{\theta_{j}\right\}$ of the learned normality for hypothesis test. The submodule keeps $\left\{\theta_{j}\right\}$ of the learned normality for testing $\left\{\left(y_{tj},{\tilde{\nu }}_{tj}\right)\right\}$ of the next wafers until the learning phase updates the learned normality.

In the one-tail hypothesis test, the null hypothesis H_0_ is “the test sample sequence is normal”, which is equivalent to “the difference value is close to 0”. The alternative hypothesis H_1_ is “the test sample sequence is abnormal”, which is equivalent to “the difference value is far larger than 0”. Under the assumption that difference values of normal sequences follow a normal distribution, the one-tail hypothesis test submodule will reject H_0_ when the difference value of the tested sample sequence is higher than the test threshold $\theta_{j}$ (Equation (13)). The output test label of SVID j of the wafer for test will thus be “abnormal” if $d_{tj}>\theta_{j}$, and “normal” if $d_{tj}\leq \theta_{j}$.

### 5.2. Functionality Evaluation

To preliminarily evaluate computational time and effectiveness in detecting abnormal sequences by the Testing phase, we use the same dataset, settings, and evaluation metrics as described in Section 4.2.1, Section 4.2.2 and Section 4.2.3 The CSST data sequence for test is known abnormal.

#### 5.2.1. Expected Results

We expect the testing phase finish its computation in just seconds since the SAE testing basically performs matrix multiplications and function substitutions. We expect a tested abnormal sequence has $\alpha_{tj}>\alpha$, i.e., its identification factor is larger than α standard deviation from the mean, say α = 3. We thus expect the testing phase to be effective.

#### 5.2.2. Evaluation Results and Discussions

The computation time of the testing phase over 20 cycles of the single SVID test data is 0.00204 s for time series and 0.00199 s for frequency spectra. This demonstrates that the testing phase can finish its computation in few seconds for hundreds SVIDs of a wafer. It is efficient enough for practical applications since CSSTs come in once a wafer finishes the recipe, which is 2 or 3 min in general.

Figure 9a depicts the difference value for an abnormal sequence in time series. The left part shows the abnormal sequence (in blue) and the right part the decoded sequence (in orange) as well as the blue sequence. Significant differences between two sequences can be seen in the right part. For example, in time period of 50 to 80 s, the abnormal sequence stays around 2900, while the decoded sequence stays around 3200. The difference value of this cycle is $d_{tj}=45,073.648$, while the learned normality has $\mu_{j}=1276.024$ and $\sigma_{j}=1732.268$. Thus $\alpha_{tj}$ is about 25.283, which is 7.4 times larger than 3. This demonstrates a successful anomaly detection.

Figure 9b depicts the difference value for an abnormal sequence on frequency spectra. Low frequency components (under 0.02 Hz) dominate both spectra, which makes differences at other frequencies relatively minor. Although the difference between two spectra are not easily identified visually, the computed difference is significant, where $d_{tj}=110.264$, and the learned normality has $\mu_{j}=10.177$, $\sigma_{j}=16.427$. Thus $\alpha_{tj}$ is about 6.093 according to Equation (14), which doubles the value 3. Again, this demonstrates that the testing phase can successfully detect this abnormal sequence in frequency spectra.

## 6. Correctness, Efficiency, and Robustness

This section studies the correctness, efficiency and robustness of STALAD. In Figure 4, STALAD provides detection results and the computation time in the equipment health dashboard from which engineers validate the correctness of the detections. Both the performance measures of correctness and efficiency are used to evaluate the ability of STALAD to deal with the challenges to identify anomaly from normal and high-rate ESD as stated in Section 2.2.

We then explore the robustness (tolerance to data) of STALAD to evaluate the effects of abnormal sequences that may appear in training and/or validation data of STALAD. As we have only limited knowledge about normal versus abnormal ESD, we may have no idea whether abnormal sequences appear in training or validation data. Note that, in Figure 4, an unsupervised learning scheme is adopted in the learning phase. The inputs to the unsupervised normal feature learning module could include biased data of abnormal sequences in historical CSST, which may lead to biased SAE weights and test thresholds, and further the biased AD results.

### 6.1. Anomaly Detection Correctness

Since engineers know the features of normal and abnormal sequences in the dataset described in Section 4.2.1, we use the same dataset to test the correctness of anomaly detection. In the dataset, its ESD of 7th and 8th SVIDs show obvious deviations from normal sequences, which are quite typical indicators to detect anomalies in an HDP-CVD tool. Engineers could easily validate the detection results of this dataset.

The results of STALAD are validated by engineers after STALAD executes on the dataset for 20 testing cycles. The settings of STALAD are the same as described in Section 4.2.2. We plot comparisons of testing cycles to visualize normal versus abnormal sequences of the testing cycles. The correctness of STALAD is defined as the percentage of correctly identified cycles in the overall 20 testing cycles.

Figure 10 shows four plots of difference values against time-series and spectral cycles between the sample sequence and its decoded sequence by time and spectral SAEs (Equations (9) and (10)) from two different SVIDs. In the plots, all the x-axes represent cycle counts, and the y-axes the difference values. 

In each plot of Figure 10, the two red vertical dashed lines divide cycles into three groups. From the lower to higher cycle counts, they are groups of training, validation, and testing data, respectively. In each plot of Figure 10, a green horizontal line indicates its test threshold. Sample sequences in the testing data that exceed the test threshold will be identified as abnormal sequences. Note that in testing data of all plots, only cycles 78 and 97 exceed the test thresholds. This indicates STALAD identifies cycles 78 and 97 as abnormal, and the other 18 testing cycles as normal, regardless of the SVIDs and the aspects of analyses.

Figure 11 depicts abnormal sequences in identified abnormal time-series cycles—cycles 78 and 97, from two different SVIDs. The *x*-axis represents the sampled time, and the *y*-axis the ESD value. Cycle 79, marked as green curve, is normal and validated by engineers. We plot cycle 79 as a standard and benchmark to identify abnormal sequences. In (a) of Figure 11, cycle 78, marked in blue, rises in 5 s earlier than the normal sequence; and in (b) of Figure 11, the same cycle drops earlier than the normal sequence. Both phenomena indicated by blue arrows in Figure 11 are confirmed by engineers as anomalies. Similarly, cycle 97, marked in orange, deviates from values of the normal sequence during time from 50 to 90 s, indicated as orange arrows. Such phenomena are also confirmed by engineers as anomalies. Figure 11 demonstrates the use of time series analysis of STALAD to correctly identify the 2 abnormal sequences among the 20 testing sample sequences.

Figure 12 depicts normal sequences in cycles 79 to 96 from two SVIDs. We plot cycle 79, marked in green, as a standard and benchmark. Cycles 80 to 96 that are confirmed by engineers as normal are all marked in orange. Note that almost all the trajectories in orange fit the green one, which implies the normal sequences are all similar. Figure 12 demonstrates the ability of time series analysis of STALAD to correctly identify these 18 normal sequences. In summary, the time series analysis of STALAD achieves 100% of AD correctness in 20 test cycles.

Figure 13 depicts abnormal sequences from 2 SVIDs. The *x*-axis represents the frequency, and the *y*-axis the magnitude. We plot cycle 79, marked in green, as a standard normal sequence comparing to abnormal sequences. Note that cycles 78 and 97, marked in blue and orange curves, show different fluctuations of magnitude from the normal sequence, especially at the part of frequencies at 0.15 Hz and lower, as shown in the red boxes in the figure. This implies that the spectral analysis of STALAD correctly identifies two abnormal sequences in 20 testing cycles with observations on obvious magnitude fluctuations at low frequency of 0.15 Hz and below.

Figure 14 depicts normal sequences from two SVIDs. We plot cycle 79, marked in green, as a standard and benchmark comparing to other normal sequences. Cycles 80 to 96 that are confirmed by engineers as normal are all marked in orange. Note that almost all the trajectories in orange fit the green one, especially at low frequencies of 0.15 Hz and below, which implies the spectral analysis of STALAD correctly identifies the 18 normal sequences among 20 testing cycles.

In summary, the spectral analysis of STALAD also achieves 100% of AD correctness on 20 testing cycles, and anomalies are identified in magnitude fluctuations at low frequency under 0.15 Hz.

### 6.2. Anomaly Detection Efficiency

In this Section, we use a dataset which contains cycles of data from 30 SVIDs in 5 days of wafer processing. Table 6 provides a property summary of the dataset. In addition to the same settings described in Section 4.2.2, Table 7 provides the settings for the experiments. As in real fab operations, there are tens to hundreds of SVIDs sensed and reported from a tool, this dataset is more realistic for study of efficiency.

Table 8 shows the statistics of computation times of STALAD on this dataset. As shown in the table, in the learning phase, it takes 1586 + 1331 = 2917 s (or 48.6 min) for both time and spectral SAEs. This is equivalent to about 16 wafers of processing, provided that each wafer cycle takes 3 min to process. On the other hand, in the testing phase, it takes 6.159 + 5.858 = 12.017 s for both time and spectral SAEs of 140 testing wafers. That is, each testing needs only 0.09 s (=12.017/140) in average, which is easy to handle the data inflow of 30 ESD points per second. Therefore, our STALAD can complete normality learning within processing time of 16 wafers and achieve real-time testing.

Note that in Table 8, the average learning time for spectral SAE (44.361 s) is 17% shorter than that for time SAE (52.852 s). This is because a spectral SAE has two less layers than a time SAE, and frequency transformation is relatively fast. Therefore, training of spectral SAEs is faster than training of time SAEs, which implies a benefit of using spectral analysis against traditional time series analysis.

Figure 15 depicts the difference values of each time series cycle in this dataset, which can be used to estimate the detection speed of STALAD. Note that the difference values significantly increase from 25 to 150 after 600th cycle, which is exactly the time when the anomaly happened. The difference values stay much higher than the threshold after cycle 610, which indicates STALAD can detect the anomaly in about 10 cycles. Given that a cycle takes 189 s to process, 10 cycles are equivalent to 31.5 min of processing time. That is, the detection speed of STALAD is 31.5 min, which is relatively fast for most of the operational fabs.

Figure 16 depicts the same dataset from the view of control wafer monitoring, which reveals potential benefits resulting from the fast computation time and detection speed of STALAD. The red line depicts the threshold, where the tool is considered abnormal when the readings are above the threshold. The green box ranged from days 20 to 25 shows the corresponding time of the dataset described in Table 6, where the tool went through an anomaly event. Note that there is another anomaly happening around day 38 while no any readings during days 25 to 38. Based on the interpolation of readings of days 25 and 38, the tool may have already shown a trend going towards abnormality before day 30. Given that STALAD can detect the normality change in half an hour, STALAD may help engineers figure out the anomaly at day 30. This indicates that STALAD may achieve up to two weeks advancement to AD complementary to the control wafer reading monitoring, which could have significant potentials for both process control and yield improvement.

### 6.3. Robustness

In this section, we discuss the effects of abnormal cycles of data contained in training and/or validation data with the analysis of the effects to test thresholds and the ability of identifying anomalies. We use the same dataset described by Table 6 due to its known boundary between normal and abnormal data. In specific, cycles before cycle 600 are normal, and cycles after cycle 600 abnormal. Among SVIDs, time series cycles of 28th SVID and spectral cycles of 29th SVID show significant changes around cycle 600 in their plots of difference values versus cycles. These provide a good fit since we can focus around cycles 600 to simulate the case that the data combine normal and abnormal sequences.

We pick various periods of the dataset to study effects of abnormality ratios in training and validation data. The *abnormality ratio* of data is defined as the percentage of the number of abnormal sequences in the data. To preserve the process order among cycles, the end cycles of periods are always cycle 700, while the start cycles of periods can vary. After the period is picked, earlier data of 60% of cycles are selected as the training data, intermediate data of 20% as the validation data, and remaining data of 20% as the testing data. We consider 10 cases of various abnormality ratios in training and validation data. All other experiment settings are the same as those in Table 7. Abnormality ratios in testing data of all experiment cases are set to 100%. We estimate the effect to the ability of STALAD by the *anomaly ratio*, which is defined as the percentage of the number of cycles in the testing data exceeding the test threshold.

Since STALAD relies on unsupervised learning to extract normality, too many abnormal sequences in training or validation data will be treated as “normal” and affect the robustness of STALAD. First, abnormal sequences in validation data will lead to high difference values and increase the test threshold. On the other hand, abnormal sequences often have high difference values away from the threshold, so the increase of the threshold may not affect testing results. Second, more than half abnormal sequences in training data will make them to dominate and reverse all the testing results. The effects should be observed from the change of anomaly ratio and test threshold as the abnormality ratio increases. We expect STALAD has some tolerance to the increase of abnormality ratio in validation data, and less than 50% toward the abnormality ratio in training data.

Table 9 shows all experiment results of 10 experiment cases. For cases 1 to 4 in Table 9, the abnormality ratios in validation data gradually increase from 0% to 33%. The time and spectral test thresholds gradually increase. The anomaly ratio, however, stays above 14.4% to 22.7%. This implies abnormal sequences in testing data often have enough high difference values. Thus the increase of the threshold doesn’t affect testing results too much. As a consequence, we conclude that time series and spectral analyses of STALAD are robust enough to the cases whose abnormality ratio in validation data is below 33%.

For cases 4 to 7 in Table 9, the abnormality ratios in validation data gradually increase from 33% to 100%. Notice that the anomaly ratios of both time series and spectral analyses of STALAD decrease linearly from about 22% to 0%, while the abnormality ratios increase from 33% to 67%. On the other hand, the test thresholds show significant increases. This coincides with our expectation where the threshold increases too much such that the testing results are affected. We conclude that the detection ability of both time series and spectral analyses of STALAD starts to decrease linearly when the abnormality ratio in validation data increases from 33% to 67%.

For cases 7 to 10 in Table 9, the abnormality ratios in training data gradually increase from 0 to 100%. Notice that the test thresholds significantly drop after the abnormality ratio is more than 50%. This implies abnormal sequences become dominant in training data, and the learned normality is “flipped”. The drop of the thresholds also leads to emergence of anomaly ratios, since now the features of abnormal sequences are learned as the representative and used to identify testing cycles. As a consequence, we conclude that STALAD starts to treat abnormal sequences as “normal” when the abnormality ratio in training data is beyond 50%.

## 7. Conclusions

This paper proposes the Spectral and Time Autoencoder Learning for Anomaly Detection (STALAD) framework to exploit unsupervised learning by Stacked AutoEncoders (SAEs) on high volume, high rate, and diversified ESD for in-line equipment anomaly detection (AD). STALAD is capable of learning normal features in data with some tolerance to abnormal data. Dynamic procedure control allows STALAD to achieve efficient testing for AD and regular learning for adaptation, which are suitable for practical fab applications. To fully utilize features of ESD and extract normal features, STALAD preprocesses ESD into cycle series and their spectral transformation (CSST), and exploits SAEs for learning salient features of CSST with some tolerance of abnormal data. To identify anomalies by the learned normality, STALAD exploits the hypothesis test on the difference values of tested ESD. ESD of an HDP-CVD tool is used as the case study to evaluate the performance of the AD correctness, tolerance to abnormal data, efficiency and adaption, and proactive benefits for fab applications. Numerical results indicate that 100% AD correctness is achieved on 20 testing wafer data, given 77 wafer data for learning. The spectral analysis gets 17% faster in computation than time series analysis. STALAD has 33% tolerance for abnormal data in the validation data when determining the hypothesis test threshold. For the case study of 30 ESD items from a real fab, STALAD is capable of performing real-time AD immediately after the completion of processing of a wafer lot. Complementary to the current practice of using control wafer monitoring for process control, the STALAD framework can gain up to 8 days’ advancement earlier for AD.

During the semiconductor equipment production cycle, an equipment often stays online to maximize its throughputs. The procedure controller in STALAD is designed to accommodate STALAD to the equipment production cycle via the initialization and training, testing, and update stages. The initialization and training stage is the only stage that cannot do anomaly testing, while it will never be entered again as long as STALAD is not shutdown. The testing stage allows STALAD doing real-time testing constantly. The update stage allows STALAD to learn new normality after some equipment events, and it doesn’t interrupt the testing stage. Therefore, STALAD can adapt to new normality while not interrupt the anomaly testing. With stages controlled by the procedure controller, STALAD could be a compatible, efficient, and adaptive AD decision support for equipment engineers.

Semiconductor manufacturing AD methods demand high reliability and accuracy. Expert knowledge from equipment engineers is often used for developing an AD method. However, existing literatures do not deal with the problems of lacks of expert knowledge in ESD. As the rapid increases in tool and process complexities, big data in both rates and volumes, ESD knowledge about their characteristics and normality will become more and more indescribable. Future work includes a theoretical approach to both performance evaluation and reliability analysis of STALAD. Error rate, false alarm ratio, and confusion matrix are some common approaches to evaluate an AD framework, while they need ground truth for evaluation. Reliability analysis without the need of ground truth has to be developed. Effects to SAE weights are left to be future work since it is hard to define a metric to quantitatively measure the change of SAE weights.

## Figures and Tables

**Figure 1 sensors-20-05650-f001:**
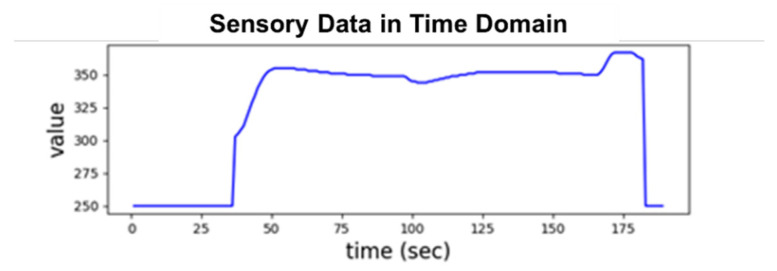
A cycle of sample sequence from an SVID collected from an HDP-CVD tool.

**Figure 2 sensors-20-05650-f002:**
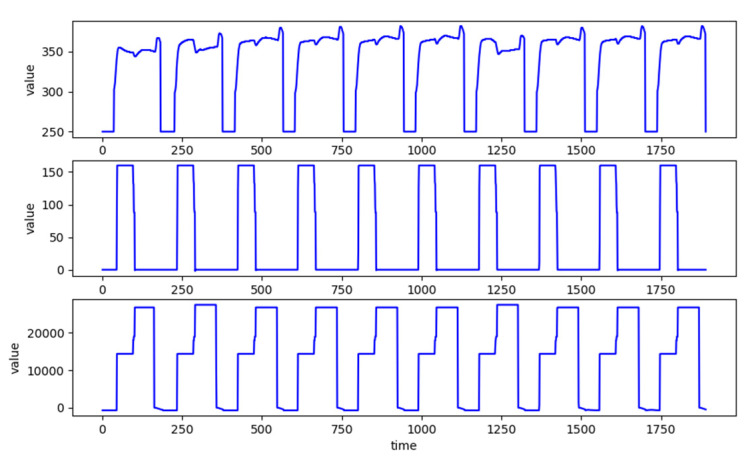
CS data containing 10 cycles from three SVIDs collected from an HDP-CVD tool.

**Figure 3 sensors-20-05650-f003:**
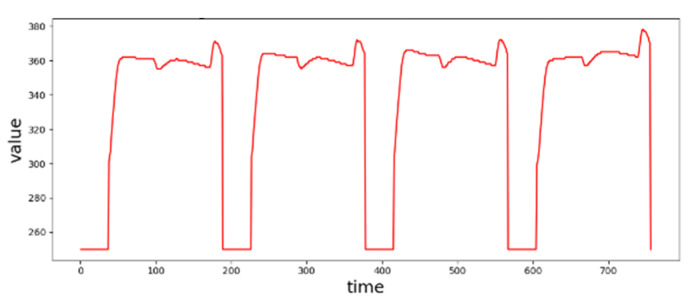
Ambiguity between normal and abnormal sequences of an SVID.

**Figure 4 sensors-20-05650-f004:**
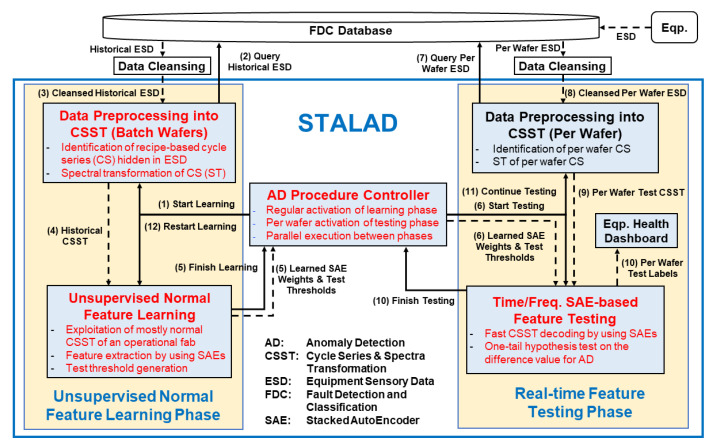
STALAD framework and operation procedure.

**Figure 5 sensors-20-05650-f005:**
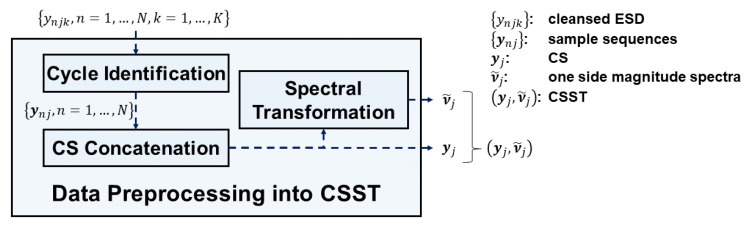
The data preprocessing module in STALAD.

**Figure 6 sensors-20-05650-f006:**
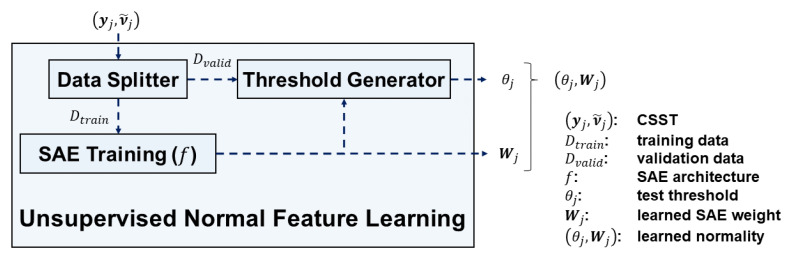
The unsupervised normal feature learning module in STALAD.

**Figure 7 sensors-20-05650-f007:**
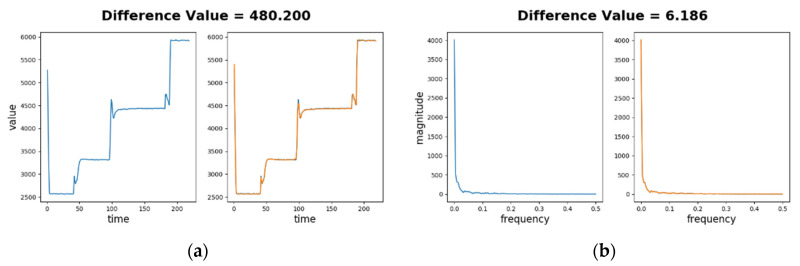
Low difference values of a normal sequence comparing to its decoded sequence: (**a**) in time series forms; (**b**) in frequency spectra forms.

**Figure 8 sensors-20-05650-f008:**
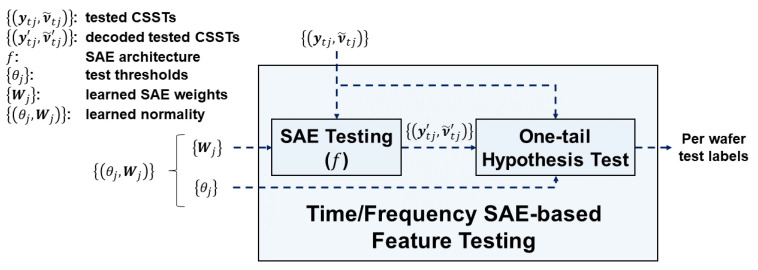
The time/frequency SAE-based feature testing module in STALAD.

**Figure 9 sensors-20-05650-f009:**
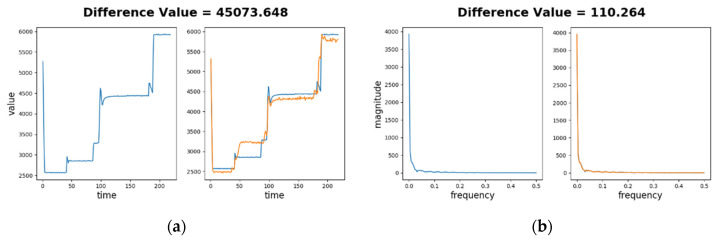
High difference values of an abnormal sequence comparing to its decoded sequence: (**a**) in time series forms; (**b**) in frequency spectra forms.

**Figure 10 sensors-20-05650-f010:**
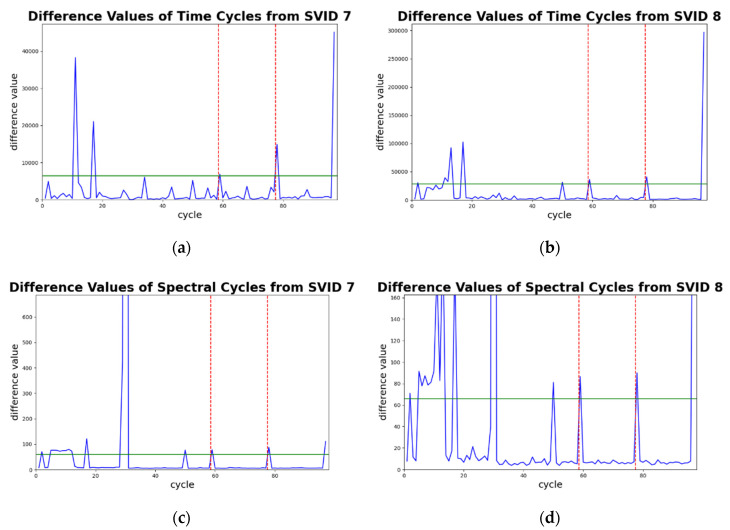
Detection of abnormal sequences by difference values from two different SVIDs: (**a**) the sample sequences of 7th SVID; (**b**) the sample sequences of 8th SVID; (**c**) ST of the sample sequences of 7th SVID; (**d**) ST of the sample sequences of 8th SVID.

**Figure 11 sensors-20-05650-f011:**
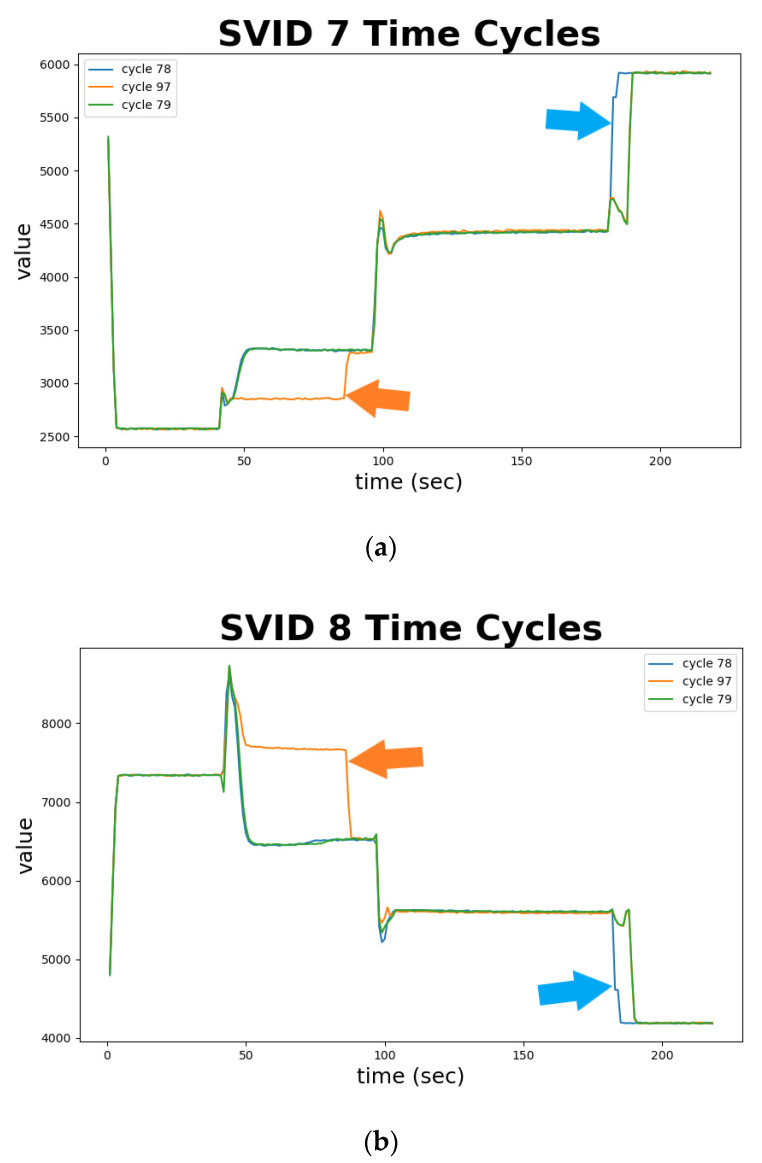
Two abnormal sequences identified against a normal sequence in time series of two SVIDs: (**a**) 7th SVID; (**b**) 8th SVID.

**Figure 12 sensors-20-05650-f012:**
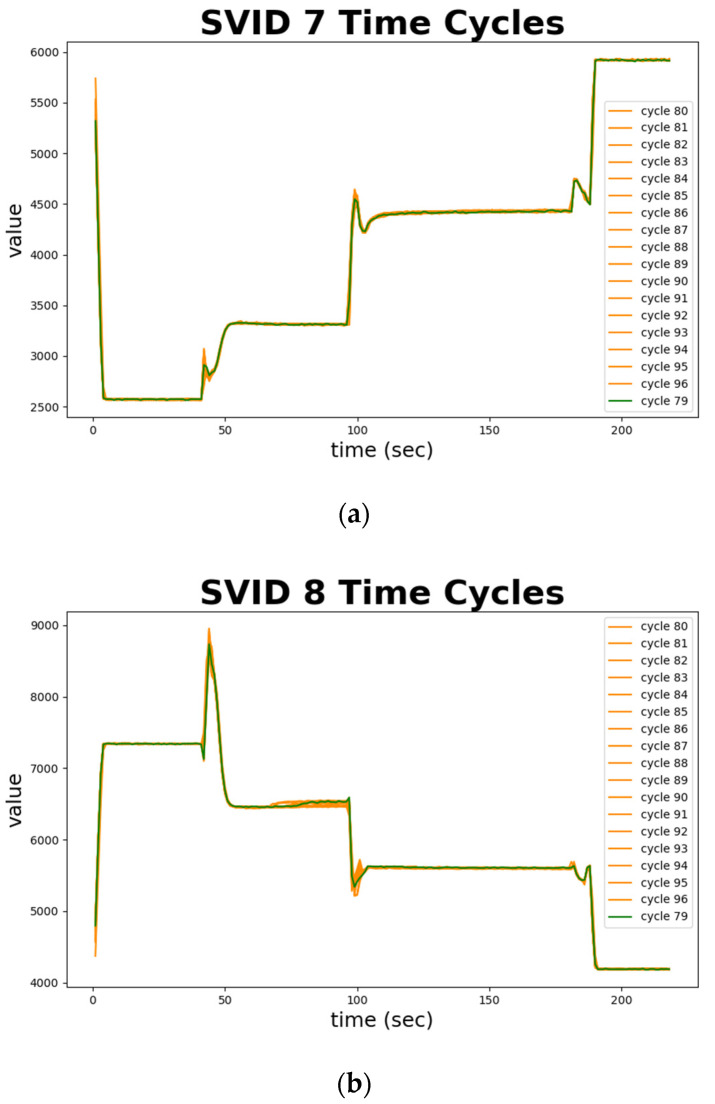
Normal sequences in time series from two SVIDs: (**a**) 7th SVID; (**b**) 8th SVID.

**Figure 13 sensors-20-05650-f013:**
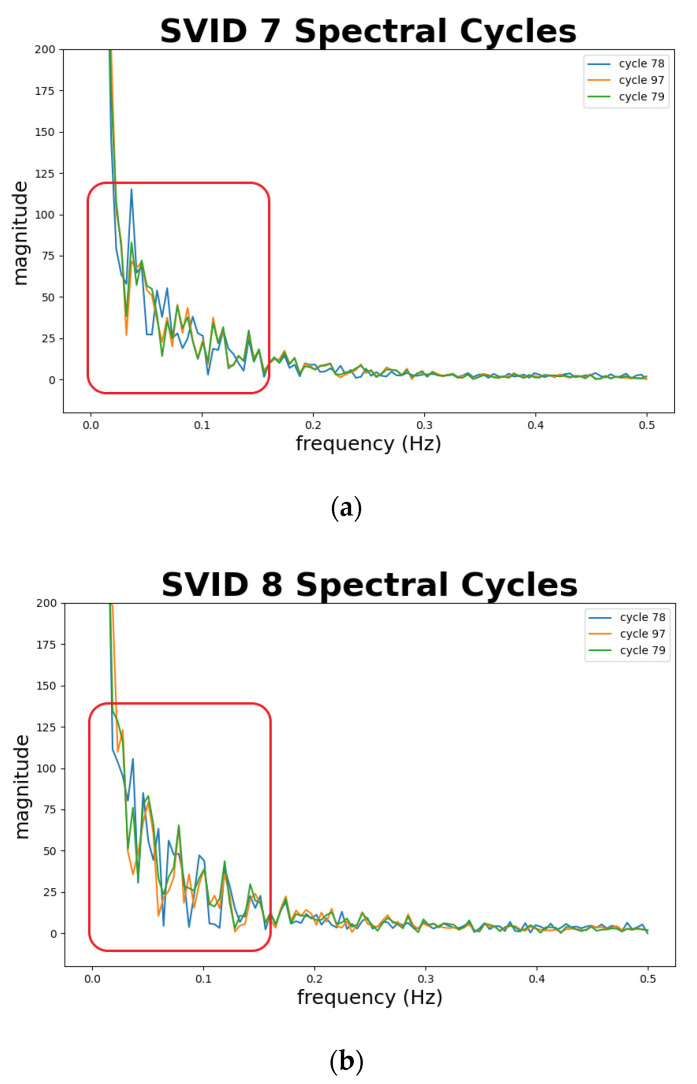
Abnormal sequences comparing to a normal sequence in spectra from two SVIDs: (**a**) 7th SVID; (**b**) 8th SVID.

**Figure 14 sensors-20-05650-f014:**
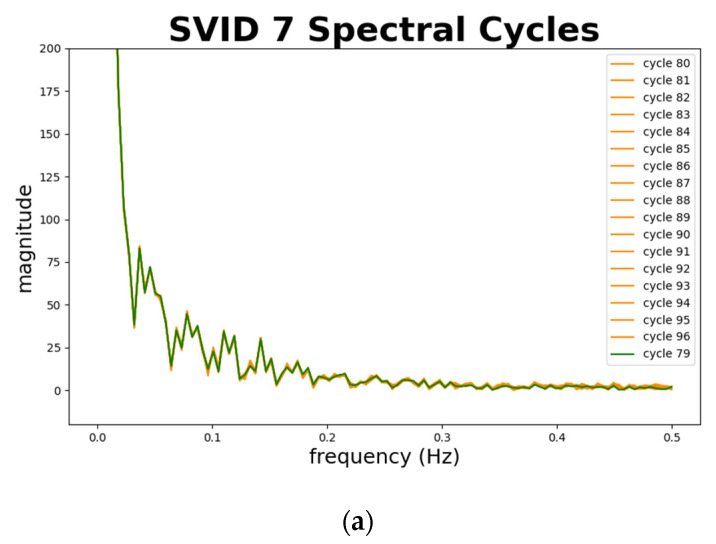
Normal sequences in spectra from 2 SVIDs: (**a**) 7th SVID; (**b**) 8th SVID.

**Figure 15 sensors-20-05650-f015:**
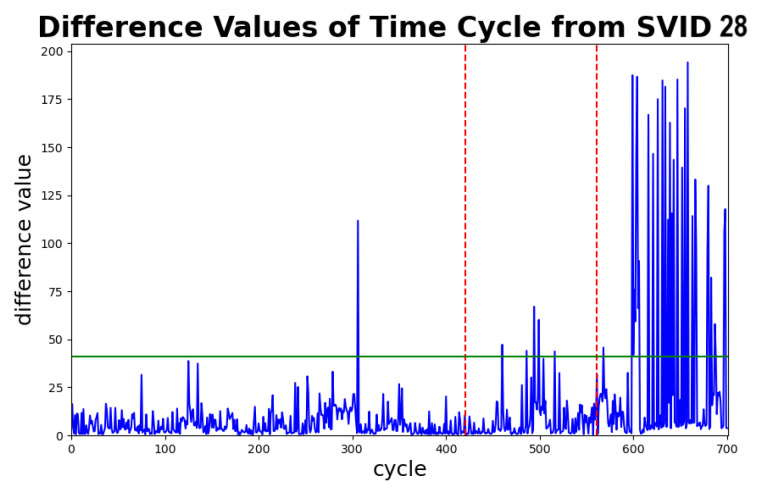
Difference values related to time series for estimating detection speed for the experiment in Section 6.2.

**Figure 16 sensors-20-05650-f016:**
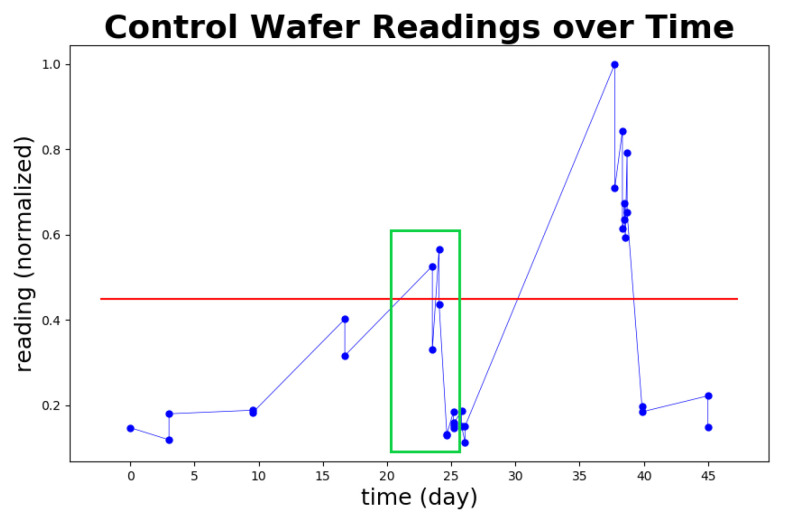
Advancement to anomaly detection by STALAD comparing to the control wafer reading monitoring for the whole anomaly event described by Table 6.

**Table 1 sensors-20-05650-t001:** Notations of constants or given sets.

Notation	Definition
N	the total number of wafers in the batch;
J	the total number of SVIDs in the batch;
K	the total number of sampling points in a cycle;
i	the imaginary unit;
r	the ratio of number of cycles in the training data, 0≤r≤1;
$D_{valid}$	the set of validation data;
$D_{train}$	the set of training data;
α	a coefficient that can tune the sensitivity of the anomaly test.

**Table 2 sensors-20-05650-t002:** Notations of functions.

**Notation**	**Definition**
ft:ℝm→ℝm	an SAE architecture for time series;
fs:ℝm→ℝm	an SAE architecture for spectra;
len(⋅):ℝm→ℤ	number of elements in a m-dimensional vector;
h(⋅):ℤ→ℤ	the dimensional reduction function, which yields the reduced dimension. h(m)=⎡m2⎤, where ⎡ ⎤ is the upper Gauss bracket;
hm(⋅):ℤ→ℤ	the iterated dimensional reduction function, i.e., hm≜h∘hm−1.

**Table 3 sensors-20-05650-t003:** Notations of indices.

Notation	Definition
n	index of a wafer, n=1,…,N;
j	index of an SVID, j=1,…,J;
k	index of a sampling point in a cycle, k=1,…,K.

**Table 4 sensors-20-05650-t004:** Notations that are expressed by indices.

Notation	Definition
$y_{njk}$	a sensor reading of SVID j of wafer n at time index k, after cleansing;
$y_{nj}$	a sample sequence of SVID j of wafer n, and $y_{nj}=\left[y_{nj1},y_{nj2},\ldots ,y_{njK}\right]$;
$y_{j}$	a CS of SVID j, and $y_{j}=\left(y_{1j},y_{2j},\ldots ,y_{Nj}\right)$;
$\nu_{nj}$	the magnitude spectrum of $y_{nj}$ after FFT;
$\nu_{njk}$	the k-th value in $\nu_{nj}$, and $\nu_{nj}=\left[\nu_{nj1},\nu_{nj2},\ldots ,\nu_{njK}\right]$;
${\tilde{\nu }}_{nj}$	the one side magnitude spectrum of $y_{nj}$;
${\tilde{\nu }}_{j}$	all one side magnitude spectra ${\tilde{\nu }}_{nj}$ of SVID j;
$\theta_{j}^{t}$	a test threshold of SVID j for time series;
$\theta_{j}^{s}$	a test threshold of SVID j for spectra;
$W_{j}^{t}$	learned SAE weights of SVID j for time series;
$W_{j}^{s}$	learned SAE weights of SVID j for spectra;
$\left(\theta_{j}^{t},W_{j}^{t}\right)$	the learned normality of SVID j for time series;
$\left(\theta_{j}^{s},W_{j}^{s}\right)$	the learned normality of SVID j for spectra.

**Table 5 sensors-20-05650-t005:** The hyperparameter settings for the experiment in Section 4.2.

Hyperparameter	Setting	Reasoning
Dimensions of each layer of time SAE	218, 109, 55, 28, 14, 7, 14, 28, 55, 109, 218	The determination of layer dimensions is described in Section 4.1. We set β=10.
Dimensions of each layer of spectral SAE	109, 55, 28, 14, 7, 14, 28, 55, 109	Same as the above.
Activation function	Leaky ReLU (leaky rate = 0.2) for all layers	Experiments show that leaky ReLU [37] can reach the lowest training error among various activation functions.
Initial weights	Normal distribution $N\left(0,\;0.1^{2}\right)$	We randomly initialize the weights, which is the usual choice described in [38].
Loss function	$\underset{n\in D_{train}}{\sum }d_{nj}$	Described in Section 4.1.
Optimizer	Adam with default settings suggested in [35]	[35] suggests Adam that is good for general problems.
Maximum training iterations	10,000	We apply the early stopping technique to prevent from overfitting. Investigations suggest that the iterations should stop around 5000 to 15,000.

**Table 6 sensors-20-05650-t006:** Dataset properties in Section 6.2.

Property	Value
# of cycles in the dataset	700
# of SVIDs in the dataset	30
# of data points in a cycle	189
Idle time between cycles	1–3 min
# of cycles collected in a day	About 140

**Table 7 sensors-20-05650-t007:** The experiment settings for the experiment in Section 6.2.

Item	Setting
# of cycles used in the training data	420
# of cycles used in the validation data	140
# of cycles used in the testing data	140
Dimensions of each layer of time SAE	189, 95, 48, 24, 12, 6, 12, 24, 48, 95, 189
Dimensions of each layer of spectral SAE	95, 48, 24, 12, 6, 12, 24, 48, 95

**Table 8 sensors-20-05650-t008:** Computation time of STALAD phases over 30 SVIDs for the experiment in Section 6.2.

Time Statistics	Learning Phase for Time SAE (s)	Testing Phase for Time SAE (s)	Learning Phase for Spectral SAE (s)	Testing Phase for Spectral SAE (s)
**Total**	1585.552	6.159	1330.818	5.858
**Average**	52.852	0.205	44.361	0.195
**Standard Deviation**	0.785	0.007	0.693	0.060

**Table 9 sensors-20-05650-t009:** The effects of abnormality ratio to the anomaly ratio and the test thresholds of time series and spectral analysis STALAD.

Case Index	Abnormality Ratio (Training)	Abnormality Ratio (Validation)	Anomaly Ratio (Time)	Test Threshold (Time)	Anomaly Ratio (Spectral)	Test Threshold (Spectral)
1	0/300 = 0%	0/100 = 0%	22/100 = 22.0%	84.56	16/100 = 16.0%	1.440
2	0/270 = 0%	10/90 = 11%	13/90 = 14.4%	39.41	15/90 = 16.7%	1.516
3	0/240 = 0%	20/80 = 25%	12/80 = 15.0%	74.07	18/80 = 22.5%	2.952
4	0/225 = 0%	25/75 = 33%	16/75 = 21.3%	104.72	17/75 = 22.7%	3.284
5	0/201 = 0%	33/67 = 49%	7/67 = 10.4%	132.12	7/67 = 10.4%	3.498
6	0/180 = 0%	40/60 = 67%	0/60 = 0.0%	168.10	0/60 = 0.0%	43.230
7	0/150 = 0%	50/50 = 100%	0/50 = 0.0%	205.68	0/50 = 0.0%	60.063
8	34/99 = 34%	33/33 = 100%	0/33 = 0.0%	438.85	0/33 = 0.0%	93.109
9	50/75 = 67%	25/25 = 100%	0/25 = 0.0%	96.30	1/25 = 4.0%	3.692
10	60/60 = 100%	20/20 = 100%	10/20 = 50.0%	14.07	1/20 = 5.0%	3.309

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
