# Peer review of "Equipment Anomaly Detection for Semiconductor Manufacturing by Exploiting Unsupervised Learning from Sensory Data†"

_sensors, 2020, doi:10.3390/s20195650_

Round 1

Reviewer 1 Report

s1. The Spectral and Time Autoencoder Learning for Anomaly Detection (STALAD) framework proposed in this paper solves the challenges of dealing with diversified and high-rate equipment sensory data (ESD) without sufficient knowledge about its normality.

s2. This paper proposes some issues and definition about anomoly detection problem exploiting time series ESD without knowledge, and solves these issues one by one in the model, which has reference significance for future problem solving.

s3. The experimental design in the thesis is clear. After the introduction of each part of the model, there are complete experiments to verify the accuracy and performance of the model.

w1. The data set used in the paper includes 97 cycles collected from 8 SVID of an HDP-CVD tool, which is used in subsequent experiments. However, on the one hand, the paper mentioned that the ESD data is growing very fast, and the data set used in the paper is not large. On the other hand, to illustrate the effectiveness of the model, using multiple sets of different data, including public data sets, etc, will be more convincing.

w2. As mentioned in the Introduction section, the author’s method is based on some design issues. And these issues are indeed worthy of research. So, what are the state-of-the-art studies on these issues, and what are the advantages or disadvantages of the methods in the paper compared with these methods. It would be better to add some related work in this paper.

w3. In the experimental part, it is better to compare the methods mentioned in the paper with the state-of-the-art work of these design issues.

Reviewer 2 Report

Dear Authors,

I have some comments on your article:

  1. Section 2.1. ESD Characteristics – please provide more details about the collected signals, something more about measuring devices and sensors
  2. All indexes in symbols in text and equations should be checked carefully.
  3. The Conclusion sections should be extended and written clearly which is the advantage of the proposed method and how to implement it for use in the production cycle.
  4. The literature review should be extended to include other methods of equipment anomaly detection for semiconductor manufacturing.
  5. Literature should be checked if there are no newer items. Especially from the last 18 months. It would be good to add several references.
